🔓 | **Open Peer Review** | Genetics and Molecular Biology | Research Article

# Sirtulin–Ypk1 regulation axis governs the TOR signaling pathway and fungal pathogenicity in *Cryptococcus neoformans*

Zhenghua Chai,[1] Yanjian Li,[2] Jing Zhang,[3] Chen Ding,[3] Xiujuan Tong,[4] Zhijie Zhang[1]

**ABSTRACT** *Cryptococcus neoformans* is a life-threatening fungal pathogen that is a causative agent for pulmonary infection and meningoencephalitis in both immunocompetent and immunodeficient individuals. Recent studies have elucidated the important function of the target of rapamycin (TOR) signaling pathway in the modulation of *C. neoformans* virulence factor production and pathogenicity in animal infection models. Herein, we discovered that Ypk1, a critical component of the TOR signaling pathway, acts as a critical modulator in fungal pathogenicity through post-translational modifications (PTMs). Mass spectrometry analysis revealed that Ypk1 is subject to protein acetylation at lysines 315 and 502, and both sites are located within kinase functional domains. Inhibition of the *C. neoformans* TOR pathway by rapamycin activates the deacetylation process for Ypk1. The *YPK1Q* strain, a hyper-acetylation of Ypk1, exhibited increased sensitivity to rapamycin, decreased capsule formation ability, reduced starvation tolerance, and diminished fungal pathogenicity, indicating that deacetylation of Ypk1 is crucial for responding to stress. Deacetylase inhibition assays have shown that sirtuin family proteins are critical to the Ypk1 deacetylation mechanism. After screening deacetylase mutants, we found that Dac1 and Dac7 directly interact with Ypk1 to facilitate the deacetylation modification process via a protein–protein interaction. These findings provide new insights into the molecular basis for regulating the TORC–Ypk1 axis and demonstrate an important function of protein acetylation in modulating fungal pathogenicity.

**IMPORTANCE** *Cryptococcus neoformans* is an important opportunistic fungal pathogen in humans. While there are currently few effective antifungal treatments, the absence of novel molecular targets in fungal pathogenicity hinders the development of new drugs. There is increasing evidence that protein post-translational modifications (PTMs) can modulate the pathogenicity of fungi. In this study, we discovered that the pathogenicity of *C. neoformans* was significantly impacted by the dynamic acetylation changes of Ypk1, the immediate downstream target of the TOR complex. We discovered that Ypk1 is acetylated at lysines 315 and 502, both of which are within kinase functional domains. Deacetylation of Ypk1 is necessary for formation of the capsule structure, the response to the TOR pathway inhibitor rapamycin, nutrient utilization, and host infection. We also demonstrate that the sirtuin protein family is involved in the Ypk1 deacetylation mechanism. We anticipate that the sirtuin–Ypk1 regulation axis could be used as a potential target for the development of antifungal medications.

**KEYWORDS** fungal infection, acetylation, *Cryptococcus neoformans*, deacetylase, fungal pathogenicity

C*ryptococcus neoformans* is an airborne opportunistic human fungal pathogen that is the causative agent of cryptococcosis in immunocompromised individuals (1, 2). It is responsible for approximately 135,900 deaths in Sub-Saharan Africa and 15%

Address correspondence to Zhijie Zhang, zzjcmu@163.com.

Zhenghua Chai and Yanjian Li contributed equally to this article. Author order was determined alphabetically.

The authors declare no conflict of interest.

See the funding table on p. 12.

of global HIV/AIDS-related deaths (3). Studies have extensively shown a correlation between protein post-translational modifications (PTMs) and pathogenicity in human fungal pathogens (4, 5). Protein lysine acetylation is a reversible PTM that is essential for fungal biology. A number of studies have shown that acetylation is critical to the modulation of stress responses, fungal fitness, and invasive growth of *Candida albicans* and *Aspergillus fumigatus* in both *in vivo* and *in vitro* environments (6–10). However, how acetylation participates in regulating the fungal pathogenicity of *C. neoformans* remains largely unknown.

Others have shown that acetylation modulation proteins such as acetyltransferases and deacetylases are critical in regulating the transcriptional responses of fungal pathogenicity factors and drug resistance machineries (11–15). Transcriptome analysis following exposure of *C. neoformans* to deacetylase inhibitors shows significant alterations in the gene expression of virulence factors (16). Furthermore, deacetylases and acetyltransferases, such as Hos1, Hda1, and Gcn5, have been shown to play essential roles in fungal pathogenicity modulations (17–26). *Cryptococcus neoformans* Gcn5, an acetyltransferase, is involved in regulating gene expression as a response to environmental stress, osmotic stress, and capsule formation (21, 25, 26). Furthermore, the histone deacetylase (HDAC) null mutant strain demonstrated a fluconazole-sensitive cell growth phenotype (27, 28). Recent analysis of the acetylome in *C. neoformans* revealed that most HDACs play essential functions in modulating *C. neoformans* virulence factors, including melanin, capsule, and fungal burdens in animals (16). Analyses of two HDACs, Dac2 (Hos2) and Dac4 (Hda1), illustrated that both are responsible for maintaining *C. neoformans* fitness and animal survival by regulating the GTP-binding domain of Tef1, the elongation factor (16).

Our previous studies have extensively mapped the acetylation sites in hundreds of *C. neoformans* orthologs of proteins involved in virulence (16). In these orthologs, Ypk1, a serine/threonine protein kinase, is reciprocally regulated by acetylation at K315 and K502. The hyperacetylation of Ypk1 defects modulates fungal fitness in lung and brain tissues. However, how these sites function in modulating the virulence factors remains uninvestigated. To further elucidate the function of Ypk1 acetylation in modulating *C. neoformans* pathogenicity, we systematically investigated the potential roles of both acetylation sites in Ypk1 protein function. We found that these sites are important modulators in the TOR process, nutrient utilization and fungal virulence. Screening *C. neoformans* deacetylase mutant strains shows that Ypk1 is a direct client protein for the sirtuins Dac1 (Sir2) and Dac7 (Hst2).

## RESULTS

### Acetylation modulates Ypk1 function in fungal pathogenicity

We have shown previously that Ypk1 acetylation is a critical part of modulating fungal fitness in lung and brain tissues (16). Mutating acetylation sites $K^{ac315}$ and $K^{ac502}$ to glutamine (Q), which mimics the hyperacetylation of Ypk1, dramatically decreased the colony-forming units in both tissues, indicating that acetylation is critical in regulating Ypk1 protein activity and fungal pathogenicity. While $K^{315}$ was localized within the predicted protein kinase domain (Fig. 1A and C), $K^{502}$ was found at the AGC kinase domain (Fig. 1B and C).

*Cryptococcus neoformans* capsule formation, used to validate the function of acetylation in Ypk1 mutant strains, showed that capsule thickness was reduced in the *ypk1Δ* strain compared to the wild-type strain (Fig. 2A). Additionally, a moderate reduction in the capsule structure was detected in the *YPK1Q* strain (expressing Ypk1$^{K315Q, K502Q}$) compared to the *YPK1R* strain (expressing Ypk1$^{K315R, K502R}$) (Fig. 2A). The evidence indicates that acetylation of Ypk1 inhibits the development of the capsular structure in *C. neoformans*, and it is highly likely that the deacetylation process is necessary for this process. Consequently, we cultured the *YPK1WT* cells in Dulbecco's modified Eagle medium (DMEM) that stimulates capsule formation and subsequently measured the level of acetylation of Ypk1 in this specific growth condition. Our analysis revealed a significant

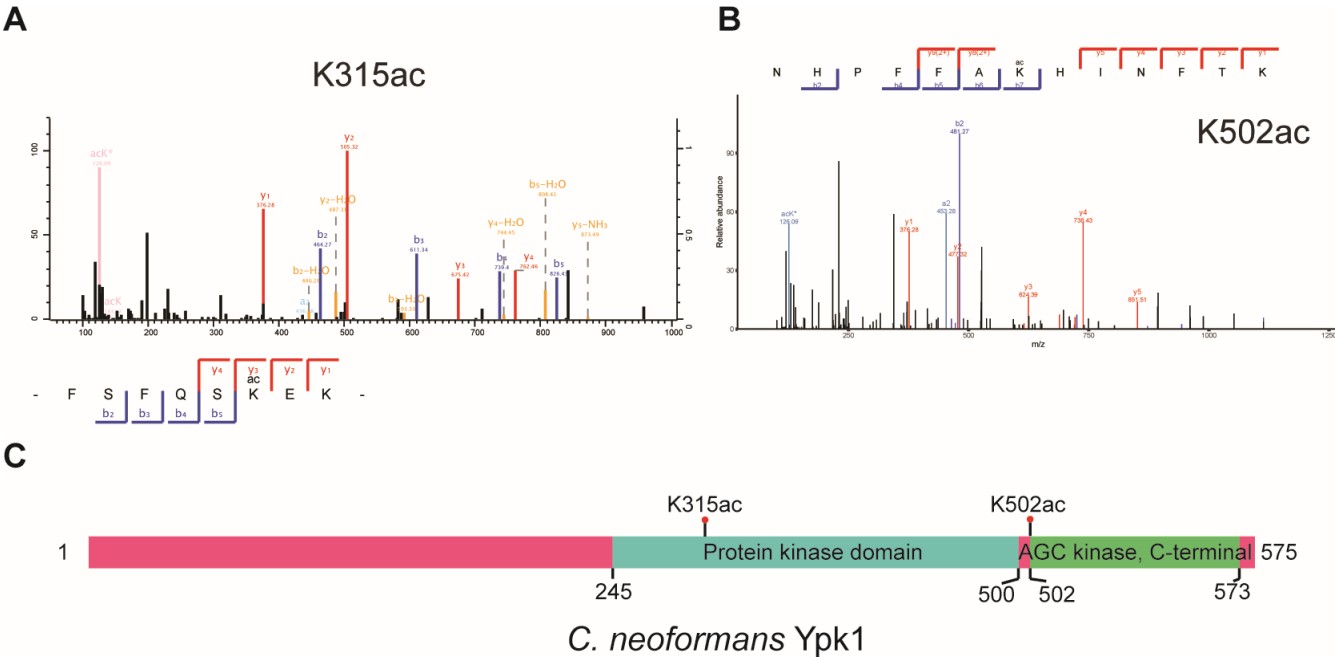

**FIG 1** *Cryptococcus neoformans* Ypk1 is an acetylated protein. (A) Acetylation mass spectrograph of Ypk1 K315. (B) Acetylation mass spectrograph of Ypk1 K502. (C) Scheme of protein domains and acetylation sites of Ypk1. Protein domains were analyzed using InterPro (https://www.ebi.ac.uk/interpro/). Acetylation sites were mapped.

reduction in the overall acetylation levels of Ypk1 when cells were cultivated in DMEM supplemented with 10% fetal bovine serum (FBS). This finding suggests that the process of capsule formation necessitates the deacetylation of Ypk1 (Fig. 2B). Consistent with this point, animal infection experiments demonstrate a prolonged animal survival rate for the *YPK1Q* strain (Fig. 2C). These data suggest that the acetylation level of Ypk1 plays an important role in modulating functional pathogenicity.

## Deacetylation is required for a Ypk1 response to rapamycin and starvation tolerance

The reciprocal BLAST analysis demonstrated significant homology between *C. neoformans* Ypk1 and *S. cerevisiae* Ypk1 (http://fungidb.org/). Studies of *S. cerevisiae* have shown that Ypk1 is a critical player in the TOR process (29–33). To examine the potential function of *C. neoformans* Ypk1 and its acetylation regulation in the TOR pathway, we challenged *ypk1* mutant strains with the TOR inhibitor, rapamycin. Consistent with the published data, we found that a loss in function of the *ypk1Δ* strain resulted in a hypersensitive cell growth phenotype when supplementing with rapamycin (Fig. 3A and B). The rapamycin-sensitive phenotype of *C. neoformans ypk1Δ* strain fully recapitulated that from *S. cerevisiae*. These data demonstrate that the function of Ypk1 is conserved across fungal species (33). Complementing a copy of the wildtype *YPK1* cDNA in the *ypk1Δ* strain (*YPK1WT*) rescued the cell growth impairment in rapamycin growth conditions. However, complementing *YPK1R* (*YPK1*[K315R; K502R] mutation) fully rescued the growth defect of the *ypk1Δ* strain. Interestingly, the *YPK1R* strain demonstrated enhanced cell growth compared to the *YPK1WT* strain (Fig. 3A and B). Additionally, complementing the *ypk1Δ* strain with *YPK1Q* (*YPK1*[K315Q; K502Q] mutation) showed reduced fungal cell growth in the presence of rapamycin, compared to the *YPK1WT* strain (Fig. 3A and B). Additional quantifiable tests were performed using liquid cell cultures, and the data fully recapitulated that obtained from YPD agar plates (Fig. 3B). These data suggest that *C. neoformans* Ypk1 functionally resembles that of *S. cerevisiae*, and acetylation negatively controls Ypk1 activity in *C. neoformans*.

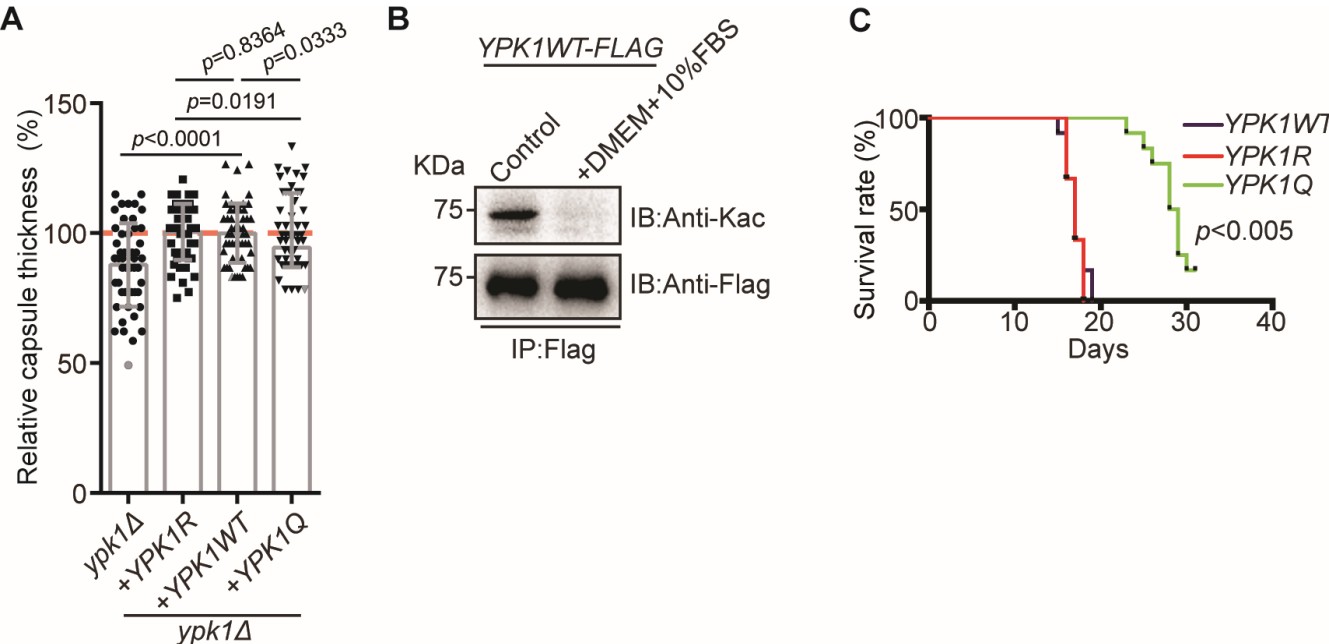

**FIG 2** Ypk1 acetylation regulates fungal pathogenicity. (A) Capsule analysis of *YPK1* mutants. Capsule thickness, measured as a relative percentages of total capsule thickness. The capsule was induced in Dulbecco's modified Eagle medium supplemented with 10% fetal calf serum for 2 days. At least 50 cells of each strain were measured. Two-tailed unpaired *t*-tests were used. (B) Immunoblots, using pan anti-Kac and anti-Flag antibodies, of proteins immunoprecipitated (using FLAG beads) from the *YPK1–FLAG* strain grown in yeast extract peptone dextrose (YPD) medium or DMEM supplemented with 10% fetal bovine serum (FBS) for 25 minutes. (C) The Kaplan–Meier survival chart of intranasally infected mice (*n* = 10 per strain).

To elucidate the potential function of acetylation in regulating Ypk1 activity, we further investigated whether rapamycin could modulate acetylation levels and the protein levels of Ypk1. To this end, we integrated a *YPK1–FLAG* plasmid in the *ypk1Δ* strain. Immunoblotting assays showed that the expression levels of the *YPK1WT–FLAG* protein in response to elevated concentrations of rapamycin were unchanged (Fig. 3C). Comparable protein levels were detected in *YPK1WT–FLAG*, *YPK1R–FLAG*, and *YPK1Q–FLAG* (Fig. 3D). Upon treating cells with rapamycin, then immunoprecipitating Ypk1WT–FLAG, and monitoring the acetylation levels, we found that exogenous rapamycin decreased the total acetylation levels of Ypk1 (Fig. 3E). Taken together, these results show that rapamycin plays no function in modulating Ypk1 protein levels, but does activate the deacetylation process of Ypk1.

In *Saccharomyces cerevisiae*, the TORC2–Ypk1 signaling is required for amino acid starvation-induced autophagy (34). Therefore, we analyzed the starvation tolerance of these mutants under nitrogen starvation (SD-N) and carbon starvation (SD-C) conditions in *C. neoformans*. Our results showed that in the 7-day nitrogen starvation treatment, the *ypk1Δ* mutant exhibited profound growth defects, as expected, and the *YPK1Q* strain exhibited growth defects, almost identical to those in the *ypk1Δ* strain. The same trend of fungal growth was also observed after the 14-day carbon starvation treatment (Fig. 3F). This phenomenon indicates that the cryptococcal Ypk1 has functional similarities to that of *S. cerevisiae*, and the level of acetylation modification of Ypk1 plays a vital regulatory role in responding to conditions of starvation.

## Deacetylation of Ypk1 is dependent on sirtuins

*Li et al*. have confirmed that there were two main deacetylase families present in *Cryptococcus neoformans*: HDAC family (histone deacetylase, including Dac2–6, 8, and 11) and sirtuin family (including Dac1, 7, and 9) (16). To elucidate the deacetylation process of Ypk1, we employed two deacetylase inhibitors: TSA, an HDAC inhibitor, and NAM, a sirtuin inhibitor. Cells treated with NAM followed by immunoprecipitation of

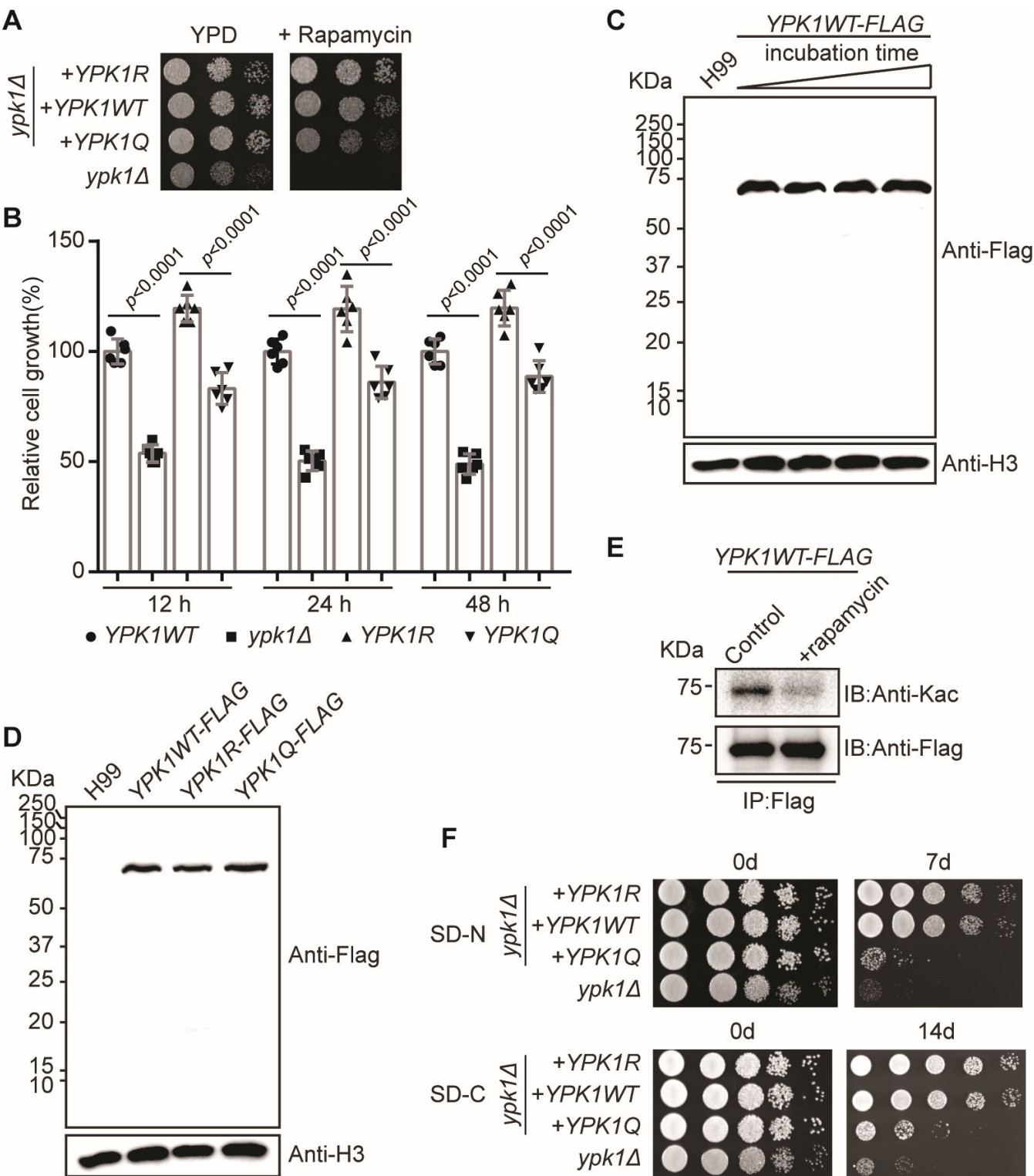

**FIG 3** Ypk1 acetylation is involved in regulating the TOR pathway. (A) Sample YPD agar plates without or with 2 ng/mL rapamycin supplementation after being separately spotted with *ypk1Δ/YPK1R*, *ypk1Δ/YPK1WT*, *ypk1Δ/YPK1Q*, and *ypk1Δ* strains followed by incubation at 30℃ for 2 days. (B) Quantification of *YPK1* mutant strains in YPD liquid cultures. Overnight fungal cultures were diluted in fresh YPD, supplemented with 2 ng/mL rapamycin. Cell densities were measured after cell growth at 12 hours, 24 hours, and 48 hours. Two-tailed unpaired *t*-tests were used. (C) Immunoblots, using anti-Flag and anti-H3 antibodies, of proteins isolated from *YPK1* mutants grown in liquid YPD media for 6 hours. (D) Immunoblots quantifying Ypk1–Flag from *YPK1WT–FLAG* grown in liquid YPD media supplemented with 2 ng/mL rapamycin at 30℃ for 0, 5, 15, or 25 minutes. Anti-H3 was used as a loading control. (E) Immunoblots, using pan anti-Kac and

**FIG 3** (Continued)

anti-Flag antibodies, of proteins immunoprecipitated (using FLAG beads) from the *YPK1–FLAG* strain grown in YPD liquid media supplemented with or without 2 ng/mL rapamycin at 30°C for 25 minutes. (F) Tolerance of strains to nitrogen or carbon starvation. *ypk1Δ/YPK1R, ypk1Δ/YPK1WT, ypk1Δ/YPK1Q*, and *ypk1Δ* strains were grown in liquid YPD for 15 hours and then grown on the SD-N medium and SD-C medium for 7 days and 14 days, respectively. Dilutions were grown on YPD plates for 2 days.

Ypk1WT-FLAG and immunoblotting assays with Kac showed a dramatic increase in the acetylation level of *C. neoformans* Ypk1; however, cells treated with TSA maintained low acetylation levels (Fig. 4A). These results suggest that Ypk1 is a client protein for sirtuins rather than HDACs. To precisely identify the corresponding deacetylases, we disrupted nine deacetylase-encoding genes in the *YPK1–FLAG* strain, but failed to detect any change in acetylation levels of Ypk1–FLAG in these strains (Fig. 4B and C).

This suggested that multiple deacetylases simultaneously regulate Ypk1 acetylation. To test this, we generated double-knockout mutants of sirtuin genes in the *YPK1–FLAG* strain and measured the acetylation level of Ypk1. Simultaneously disrupting *DAC1* and *DAC7* led to enhanced acetylation (Fig. 5A). Consistent with the rapamycin-sensitive growth phenotype of *YPK1Q, dac1Δ dac7Δ* impairs cell growth in the presence of rapamycin (Fig. 5B). Next, we examined the capsule structure of the knockout strains.

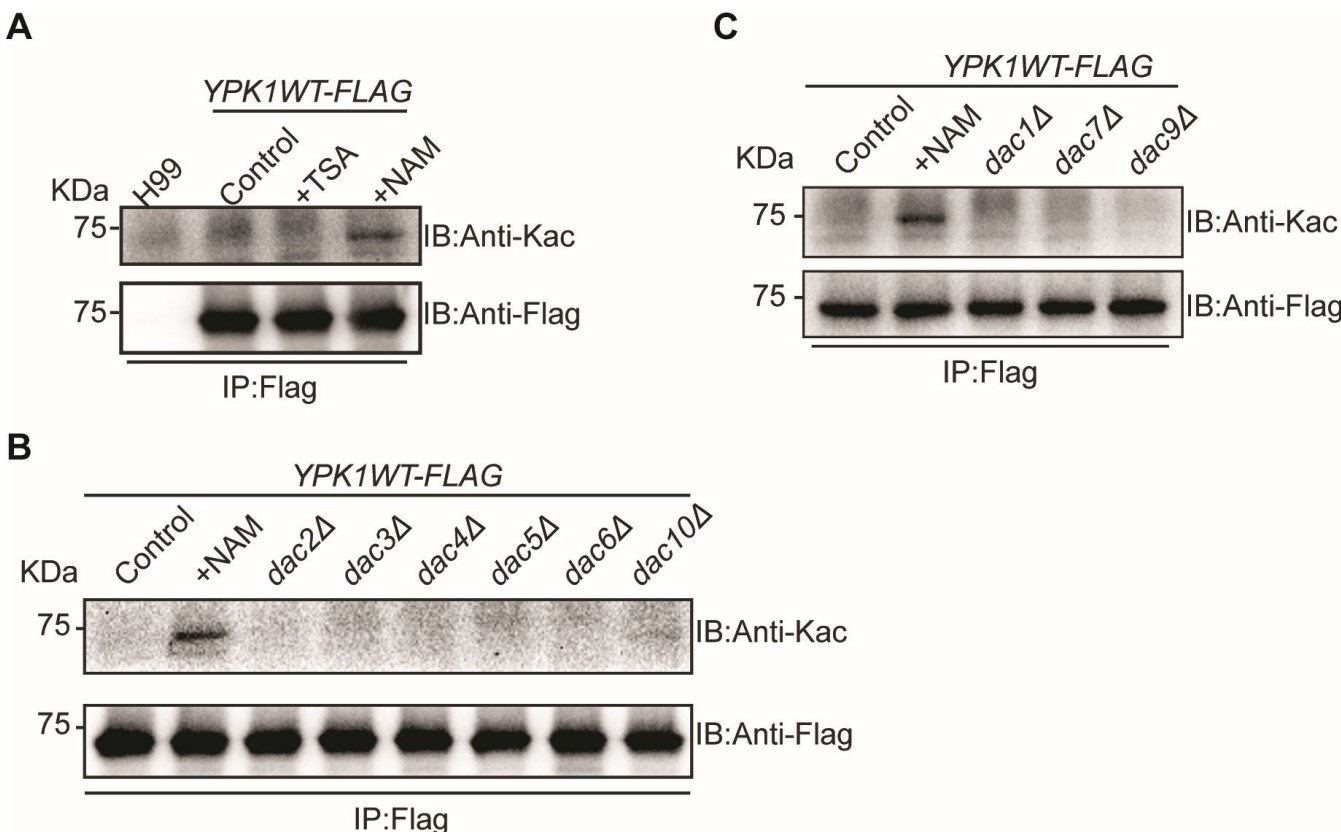

**FIG 4** The sirtuin deacetylase family regulates Ypk1 acetylation. (A) Immunoblots, using anti-Kac and anti-Flag antibodies, showing *YPK1* acetylation after growing the *YPK1–FLAG* strain in YPD media supplemented with 3 µM TSA or 20 mM NAM at 30°C for 6 hours, isolating the proteins, and then performing Flag immunoprecipitation. The control was the *YPK1–FLAG* strain grown in unaltered liquid YPD media. (B) Immunoblots, using anti-Kac and anti-Flag antibodies, showing *YPK1* acetylation after separately growing the HDAC knockout strains *dac2Δ/YPK1–FLAG, dac4Δ/YPK1–FLAG, dac5Δ/YPK1–FLAG, dac6Δ/YPK1–FLAG*, and *dac10Δ/YPK1–FLAG* in YPD media supplemented with 20 mM NAM at 30°C for 6 hours, isolating the proteins, and then performing Flag immunoprecipitation. The controls were the *YPK1–FLAG* strains grown in unaltered liquid YPD media. (C) Immunoblots, using anti-Kac and anti-Flag antibodies, showing *YPK1* acetylation after separately growing the sirtuin knockout strains *dac1Δ/YPK1–FLAG, dac7Δ/YPK1–FLAG*, and *dac9Δ/YPK1–FLAG* in YPD media supplemented with 20 mM NAM or 3 µM TSA at 30°C for 6 hours, isolating the proteins, and then performing Flag immunoprecipitation. The controls were the *YPK1–FLAG* strains grown in unaltered liquid YPD media.

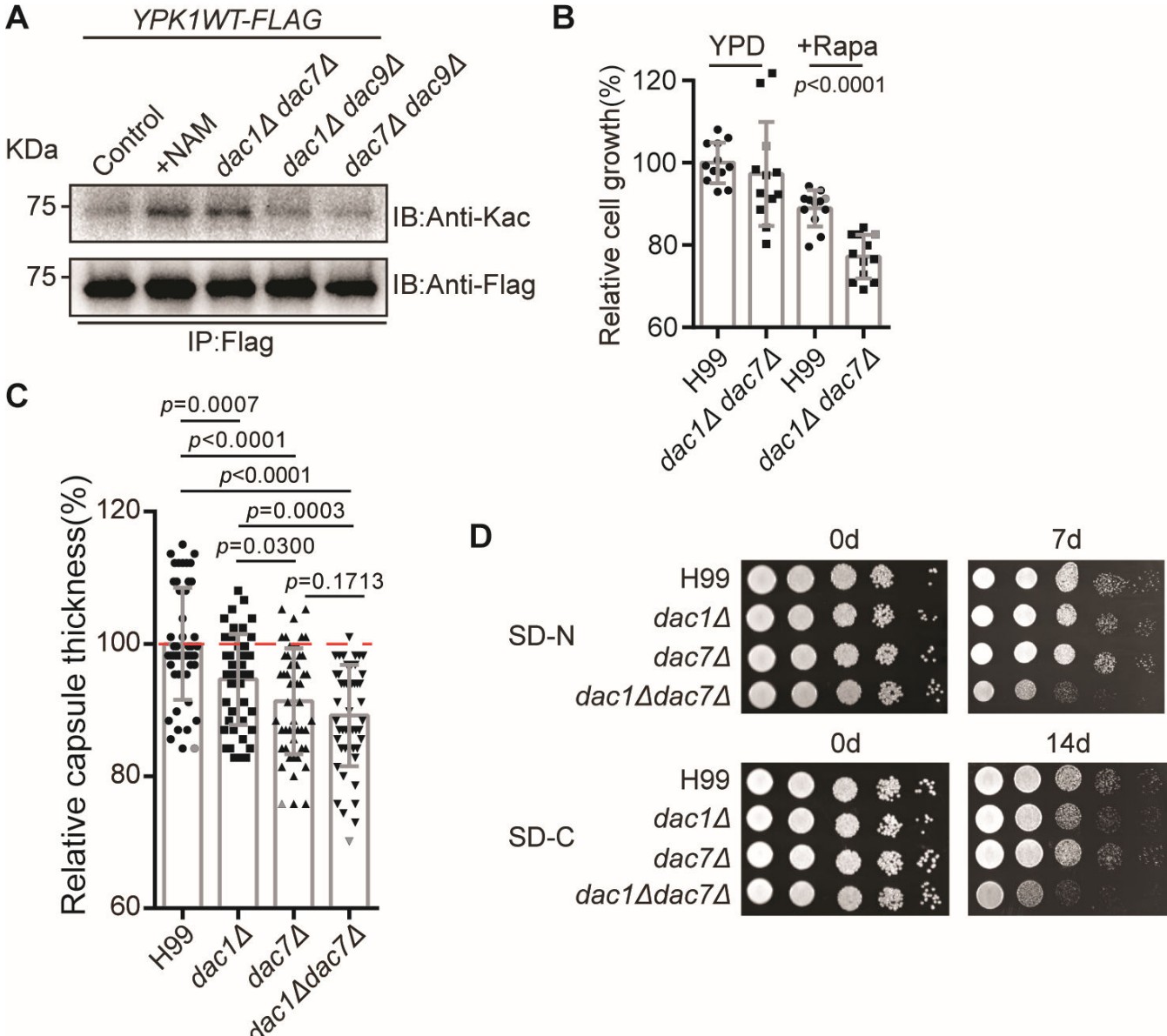

**FIG 5** Dac1 and Dac7 are critical for Ypk1 acetylation. (A) Immunoblots, using anti-Kac and anti-Flag antibodies, showing *YPK1* acetylation after separately growing the sirtuin double-knockout strains *dac1Δdac7Δ/YPK1–FLAG*, *dac7Δdac9Δ/YPK1–FLAG*, and *dac1Δdac9Δ/YPK1–FLAG* in YPD media supplemented with 20 mM NAM at 30°C for 6 hours, isolating the proteins, and then performing immunoprecipitation. The controls were *YPK1–FLAG* strains grown in unaltered liquid YPD media. (B) Immunoblots showing that cell growth in the *dac7Δ/dac9Δ* strain is sensitive to rapamycin. Left, H99 and *dac7Δ/dac9Δ* strains grown in YPD media. Right, the same grown supplemented with 2 ng/mL rapamycin. (C) Capsule analysis of knockout strains. Capsule thickness, measured as a relative percentage of total capsule thickness. The capsule was induced in Dulbecco's modified Eagle medium supplemented with 10% fetal calf serum for 2 days. At least 50 cells of each strain were measured. Two-tailed unpaired *t*-tests were used. (D) Tolerance of strains to nitrogen or carbon starvation conditions. H99, *dac1Δ*, *dac7Δ*, and *dac1Δdac7Δ* strains were grown in liquid YPD for 15 hours and then grown on SD-N medium and SD-C medium for 7 days and 14 days, respectively. Dilutions were grown on YPD plates for 2 days.

These results showed a significant reduction in the thickness of the capsule structure in all three mutant strains, especially in the *dac1Δdac7Δ* double-knockout strain (Fig. 5C). It is consistent with the phenotype of the *YPK1Q* strain's capsule structure (Fig. 2A). Furthermore, we tested the nutrient starvation tolerance of the sirtuin knockout strains. In the conditions of nitrogen and carbon starvation, the *dac1Δdac7Δ* double-knockout strain showed profound growth defects, and this was recapitulated by the growth phenotype of that from the *YPK1Q* strain (Fig. 5D). Together, these data demonstrate

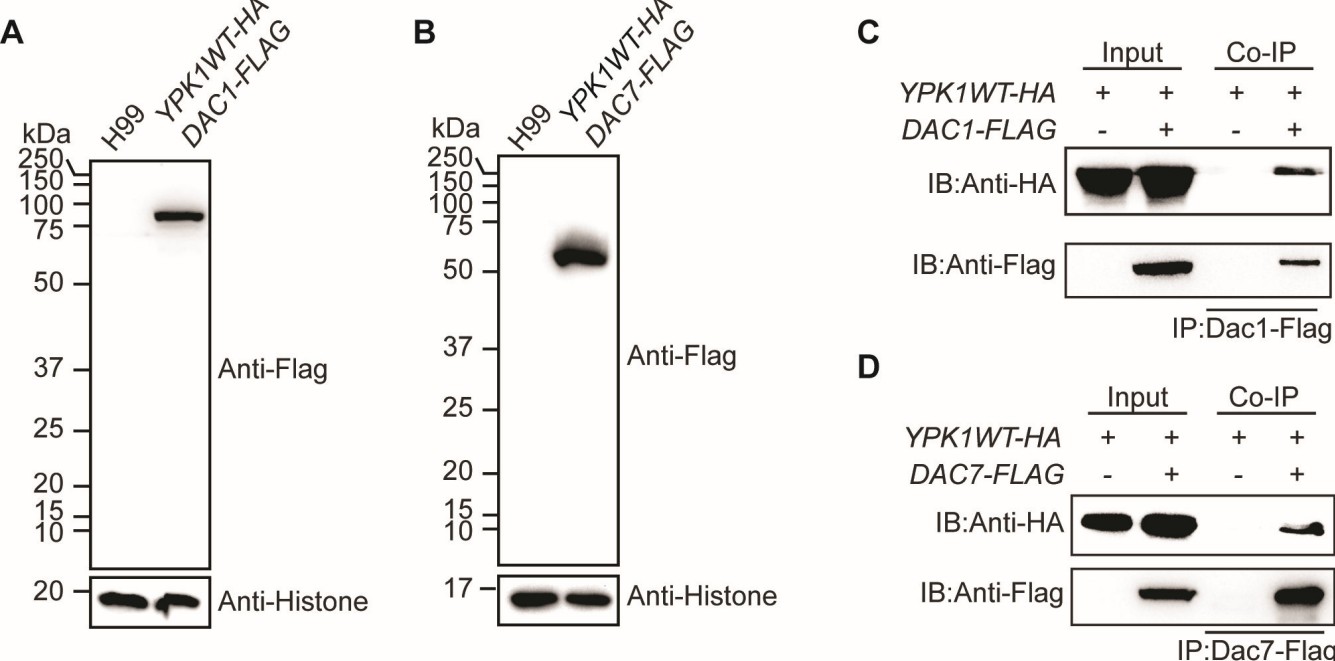

**FIG 6** Dac1 and Dac7 can interact with Ypk1. (A) Immunoblots, using anti-Flag and anti-H3 antibodies, of protein samples isolated from H99 and *YPK1–HA/ DAC1–FLAG* strains. (B) Immunoblots, using anti-Flag and anti-H3 antibodies, of protein samples isolated from H99 and *YPK1–HA/DAC7–FLAG* strains. (C) Analysis of Ypk1–HA and Dac1–Flag using a coimmunoprecipation (Co-IP) assay performed on the *YPK1–HA/DAC1–FLAG* strain. Dac1–Flag was precipitated using Flag beads, and *YPK1–HA* was detected using anti-HA antibodies. (D) Analysis of Ypk1–HA and Dac7–Flag using a Co-IP assay performed on the *YPK1–HA/DAC7–FLAG* strain. Dac7–Flag was precipitated using Flag beads, and *YPK1–HA* was detected using anti-HA antibodies.

that both Dac1 and Dac7 have overlapping functions in regulating the deacetylation of Ypk1, which is crucial for adapting the cellular response to nutrition deficiency-induced stress.

To determine whether Dac1 or Dac7 physically interact with Ypk1, two strains, *YPK1-HA/DAC1–FLAG* and *YPK1-HA/DAC7–FLAG*, were generated, and the expressions of Dac1–FLAG and Dac7–FLAG were confirmed using immunoblotting (Fig. 6A and B). We showed that both of these can be immunoprecipitated with Ypk1-HA (Fig. 6C and D). Collectively, these results show that Ypk1 activity is regulated by acetylation via two independent deacetylases, Dac1 and Dac7.

## DISCUSSION

The TOR pathway, first identified in *S. cerevisiae*, is a highly conserved process across taxa, playing essential functions in the regulation of a range of biological processes including autophagy, endocytosis, amino acid acquisition, and protein translation (34–43). While one copy of *TOR* was identified in the genomes of many organisms, two paralogs, Tor1 and Tor2, have been found in *S. cerevisiae* (44). They form two independent TOR complexes: TOR complex 1 (TORC1) and TOR complex 2 (TORC2) (44). The *S. cerevisiae* TORC2 is signally transduced via its downstream protein kinase Ypk1, which is functionally associated with sphingolipid homeostasis, mitochondrial respiration, plasma membrane integrity, and many other cellular processes (29, 30, 32, 33, 45). Two homologs of TOR were identified in the human fungal pathogen *C. neoformans*: Tor1 and Tlk1 (46). Molecular gene manipulation of *TOR1* has shown that *C. neoformans TOR1* is an essential gene, and the encoding protein product regulates numerous important biological responses (47). However, as a downstream target of Tor1, Ypk1 is indispensable for cell growth in *C. neoformans* and is required when growing in the presence of rapamycin (48). The *C. neoformans* Ypk1 has been shown to be involved in drug

resistance, and its function is critical for fungal pathogenicity in a mouse infection model when an intravenous infection route is used (48).

Studies of *S. cerevisiae* have shown that Ypk1 functionally depends on its PTM, phosphorylation. Early studies showed that Pkh1 and Pkh2 phosphorylate Thr 504 on Ypk1, that TORC2 phosphorylates Ser 644 and Thr 662, and that Fpk1 and Fpk2 phosphorylate Ser 51 and Ser 71, respectively (33, 49). More recently, the C-terminus of Ypk1 has been shown to provide four additional TORC2-dependent phosphorylation sites: Ser 653, Ser 671, Ser 672, and Ser 678 (33). In *C. neoformans,* two phosphorylation sites that were presumptively identified are Thr402, essential for Ypk1 function in fluconazole resistance, and Ser 562, indispensable for Ypk1 function (48). Further evaluation of the protein modification of *C. neoformans* Ypk1 and its functions in regulating fungal pathogenicity and drug resistance would require in-depth mass spectrometry.

In our previous study, we utilized acetylation mass spectrometry to decipher the novel post-translational modification functions of Ypk1 (16). The results showed that the null *ypk1* mutant produced an avirulence phenotype and the hyperacetylation of Ypk1, *YPK1Q* strain, showed a dramatic reduction in fungal burdens in lung and brain tissues (16).These results suggest that acetylation negatively regulates Ypk1 activity, thus hampering the protein function in modulating fungal pathogenicity in the host.

Interestingly, the *YPK1R* mutant consistently demonstrates superior growth compared to the *YPK1WT* strain in both their spotting and liquid culture assays. The primary factor responsible for this is the acetylation state of the wild-type Ypk1 protein. The Ypk1R and Ypk1Q mutagenesis approaches produce Ypk1 proteins with acetylation levels of 0% and 100%, respectively. The acetylation level of the wild-type Ypk1 protein varies between the deacetylation and complete acetylation status.

Our findings indicate that the *YPK1R* strain displays enhanced growth in different stressful situations, suggesting that deacetylation of Ypk1 is a beneficial mechanism for *C. neoformans* to cope with environmental stress. Nevertheless, the mechanism by which acetylation of Ypk1 regulates its functionality remains unknown. There are two potential scenarios. Considering that both acetylation sites are located inside the kinase domain, it is likely that acetylation has a significant role in the activity of the kinase. This, in turn, contributes to the regulation of phosphorylation for downstream targets of Ypk1. Furthermore, considering that Ypk1 undergoes both phosphorylation and acetylation, it is possible that these two changes play a role in a complex cross-talk networking mechanism, where one modification greatly influences the other. In order to gain a deeper understanding of the relationship between these two alterations, it is necessary to conduct a comprehensive examination of Ypk1 phosphorylation and its corresponding target proteins by the utilization of mass spectrometry.

The addition of rapamycin reduced the overall acetylation level of Ypk1. This indicates that *Cryptococcus* maintains a balance between inhibiting the TOR signaling transduction pathway and entering the deacetylation process of Ypk1, which is regulated by the sirtuin family. We have demonstrated that the deacetylation of Ypk1 is dependent on both Dac1 and Dac7, which are members of the sirtuin family of proteins, through a protein–protein interaction. The Ypk1 protein activity is associated with stress tolerance under nutritional deprivation conditions, as supported by the rapamycin treatment results and the published data on *S. cerevisiae*.

In conclusion, this study demonstrates a new PTM process of Ypk1, an essential player in the TOR process. Acetylation is critical for Ypk1 protein function in fungal virulence production and the pathogenicity machinery, and this study shows that not only is this a negative regulator for Ypk1, but also that deacetylation proteins are required for revealing the Ypk1 protein function. Given the essential function of Ypk1 in regulating the TOR process and fungal pathogenicity, this work enlightens the development of a novel drug target for fungal infection therapy.

## MATERIALS AND METHODS

### Strain and media

*Cryptococcus neoformans* wild-type strain H99 and mutant strains were routinely grown in liquid YPD media or YPD agar (1% yeast extract, 2% peptone, 2% dextrose, with or without 2% agar). To conduct the rapamycin test, rapamycin was added to liquid YPD media or YPD agar, reaching a final concentration of 2 ng/mL. To conduct an acetylation analysis, a deacetylase inhibitor, either trichostatin A (TSA) or nicotinamide (NAM) was added, reaching a final concentration of 3 µM or 20 mM, respectively. Using nitrogen-starvation medium [SD-N; 0.17% yeast nitrogen base (without amino acids and ammonium sulfate) and 2% glucose] and carbon-starvation medium [SD-C; 0.17% yeast nitrogen base (without amino acids and ammonium sulfate), 0.5% ammonium sulfate, 0.5% casamino acids, 0.002% tryptophan, 0.002% adenine, and 0.002% uracil) for nitrogen and carbon starvation treatments, respectively (50). Nourseothricin (100 µg/mL) and neomycin (G418) (200 µg/mL) were used to generate mutants. Dulbecco's modified Eagle medium supplemented with 10% fetal calf serum (FCS) was used for capsule formation.

### Strain generation

*C. neoformans* mutants were generated using the H99 strain and biolistic transformation (51). The *R* and *Q* mutants of *YPK1*WT were provided by Ding's Lab (16). To generate the *YPK1–FLAG dac1Δ dac7Δ* strain, *DAC1* and *DAC7* were sequentially disrupted in the *YPK1–FLAG* strain. The upstream DNA sequences of *DAC1* and *DAC7* were amplified using the primer pairs 2,383/2,384 and 2,407/2,408, respectively. The downstream DNA sequences of *DAC1* and *DAC7* were amplified using the primer pairs 2,385/2,386 and 2,409/2,410, respectively. Overlapping PCRs were performed to join the upstream sequences, selected markers, and downstream sequences and to generate the integrated knockout cassettes.

The strains *YPK1–HA DAC1–FLAG* and *YPK1HADAC7–FLAG* were generated and used for co-immunoprecipitation assays. The *YPK1–HA* strain was first generated using the plasmid *pHYG* (containing a hygromycin resistance marker under the control of the *Cryptococcus GPD* promoter). The cDNA sequence of *YPK1* was amplified using the primers 3,767 and 3,768, introducing the restriction sites *Not*I and *Stu*I, respectively. The digested PCR fragment was then cloned into the *pHYG* plasmid, and the resulting plasmid was transformed into the H99 strain. Expression of Ypk1HA was confirmed by immunoblotting with the anti-HA antibody. The cDNA sequences of *DAC1* and *DAC7* were amplified using the primer pairs 3,667/3,510 and 3,670/3,671, respectively. Overlapping PCRs were performed to generate upstream-*TEF1p–DAC1–4XFLAG–NAT*-downstream and upstream-*TEF1p–DAC7–4XFLAG–NAT*-downstream, which were introduced into the *YPK1HA* strain. The protein expressions of Dac1–Flag and Dac7–Flag were confirmed using anti-Flag antibody.

### Starvation survival assays

Overnight cultures of *C. neoformans* were harvested, washed three times with sterile PBS, and resuspended in 5 mL of SD-N or SD-C to a final concentration of $1 \times 10^7$ cells/mL. The cultures were incubated with shaking at 30°C and 200 rpm for 7 days to induce nitrogen starvation and 14 days to induce carbon starvation. Subsequently, serially diluted cell suspensions at a starting concentration of $1 \times 10^7$ cells/mL were spotted on YPD agar plates. Plates were then incubated at 30°C for 2 days to determine the survival rate under starvation conditions.

### Animal infection

Six- to 8-week-old female mice (C57BL/6) were purchased from Changsheng Biotech (Liaoning, China) and used for infection experiments. Separately, each *C. neoformans* strain was suspended ($10^5$ fungal cells in 50-µL PBS buffer) and intranasally infected in

ten mice (52). Mouse care and infection took place at the College of Life and Health Sciences of Northeastern University under an alternating 12-hour light–dark cycle and *ad libitum* access to food and water. Infected mice were weighed 10 days after infection and then the morbidity was monitored twice daily. Mice were sacrificed at the end of the experiment.

## Capsule formation

Capsule release experiments were performed as described elsewhere (53). Briefly, cells were grown overnight in YPD medium and then washed three times with PBS. The cells were resuspended in Dulbecco's modified Eagle medium supplemented with 10% fetal calf serum and then incubated for 3 days at 37°C in 5% $CO_2$. To observe capsule size, a Leica optical microscope (DM 2500) was used to capture images of cells (generally hundreds) stained with India ink. The size of the surrounding capsule was measured. The data were normalized using the thickness of the *YPK1WT* strain (in Fig. 2A) and H99 (in Fig. 5C) as the standard.

## Cell growth curves

In this study, 96-well microtitre plates were used to assess the growth of cells in liquid culture medium. Overnight YPD cultures were washed three times with phosphate-buffered saline (PBS) and diluted to an optical density of 0.02 at 600 nm in fresh YPD, supplemented with 2 ng/mL rapamycin. Following this, 200 µL of the resulting cell suspension was carefully dispensed into individual wells of a 96-well plate. The well plate was subjected to incubation at a temperature of 30°C for either 12, 24, or 48 hours. Subsequently, optical density measurements at a wavelength of 600 nm were obtained using a Synergy HTX microplate reader manufactured by BioTek. The growth of the relevant strain was standardized by normalizing it to the well without rapamycin treatment. Six or twelve biological replicates were conducted for each strain. The data were graphed utilizing GraphPad Prism software. Two-tailed unpaired *t*-tests were used.

## Protein immunoprecipitation and immunoblotting analysis

Protein immunoprecipitation and immunoblotting analysis were performed as described elsewhere (16). Briefly, protein samples were isolated using a Mini-Beadbeater-16 (BioSpec) and lysis buffer (50 mM Tris-HCl, 150 mM NaCl, 0.1% NP-40, pH 7.5) supplemented with 1X protease inhibitor cocktail (CWBio), 40 mM PMSF, 3 µM TSA, and 20 mM NAM. Anti-FLAG M2 magnetic beads (Sigma) were used in the protein IP. For detection of protein acetylation, after incubating the protein samples with Anti-FLAG M2 magnetic beads for 4 hours, we washed them four times with TBS (50 mM Tris-HCl, 150 mM NaCl, 1% Triton X-100, pH 7.4) supplemented with 3 µM TSA and 20 mM NAM. Then, the bound proteins were extracted into the protein loading buffer at 95°C for 5 min, followed by Western blotting. Monoclonal and polyclonal Kac (1:2500, PTM Bio) was used to detect the level of protein acetylation. Protein–protein interactions were detected after washing with TBS buffer without Triton X-100. The bound proteins were extracted into the protein loading buffer at 95°C for 5 min, followed by Western blotting. Anti-M2 Flag (1:5000, Abcam), anti-HA (1:5000, Abcam), and anti-Histone H3 (1:5000, Cell Signaling Technology) were used for detection of the target protein. The signal was captured using a ChemiDoc XRS+ (Bio-Rad).

## Mass spectrometry

Mass spectrometry was performed as described elsewhere (16). Overnight cultures of H99 were diluted in fresh YPD media and incubated at 30°C or 37°C for 6 hours. *Cndac2Δ* and *Cndac4Δ* cells were cultured at 30°C, and cells in the exponential phase were used. The protein was extracted using acetylome lysis buffer (8 M urea, 1% Triton X-100, 10 mM dithiothreitol, 1% protease inhibitor cocktail, 3 µM TSA, 50 mM NAM, and 2 mM EDTA), then it was treated with dithiothreitol (5 mM) and iodoacetamide (11 mM),

diluted, and digested with trypsin at 37°C overnight. The resulting digested protein samples were desalted and vacuum-dried, then the peptides were resuspended in 0.5 M TEAB, and 6-plex TMT labeling was performed. High-performance liquid chromatography fractionation was performed, and the enriched acetylated peptides were isolated by immunoaffinity precipitation. The label-free peptides and TMT-labeled peptides were analyzed using LC–MS/MS, and the collected MS/MS data were processed using the MaxQuant search engine (version 1.5.2.8).

## Statistics and reproducibility

The log-rank (Mantel–Cox) statistical test was used to analyze these animal survival tests. Paired-sample $t$ tests were used in two-sample comparisons. Actual $P$-values were used consistently for representation.

## ACKNOWLEDGMENTS

This research was supported by the National Natural Science Foundation of China (31870140 and 32270205 to C.D.).

Y.L., Z.Z., and C.D. designed the study. Z.C. and Y.L. performed the experiments. Z.C., Y.L., Z.Z., and C.D. analyzed the data. C.D. and Z.Z. provided equipment and funding. Y.L., Z.Z., and C.D. wrote the manuscript.

## AUTHOR AFFILIATIONS

[1]Department of Laboratory Medicine of Shengjing Hospital of China Medical University, Shenyang, China

[2]College of Sciences, Northeastern University, Shenyang, China

[3]College of Life and Health Sciences, Northeastern University, Shenyang, China

[4]Department of Laboratory Medicine of Central Hospital of Chaoyang, Chaoyang, China

## AUTHOR ORCIDs

Yanjian Li  http://orcid.org/0009-0008-1525-2223
Zhijie Zhang  http://orcid.org/0000-0002-9727-7043

## FUNDING

| Funder | Grant(s) | Author(s) |
|---|---|---|
| MOST | National Natural Science Foundation of China (NSFC) | 31870140, 32270205 | Chen Ding |

## AUTHOR CONTRIBUTIONS

Zhenghua Chai, Conceptualization, Data curation, Methodology | Yanjian Li, Conceptualization, Data curation, Formal analysis | Jing Zhang, Conceptualization, Data curation, Formal analysis | Chen Ding, Conceptualization, Funding acquisition, Investigation, Writing – original draft, Writing – review and editing | Xiujuan Tong, Conceptualization | Zhijie Zhang, Conceptualization, Data curation, Methodology, Writing – original draft, Writing – review and editing

## DATA AVAILABILITY

The authors confirm that the data supporting the findings of this study are available within the article and supplemental material.

## ETHICS APPROVAL

All animal experiments were reviewed and approved by the Research Ethics Committees at the College of Life and Health Sciences of Northeastern University (approval no. 16099M).

## ADDITIONAL FILES

The following material is available online.

### Supplemental Material

**Fig. S1 (Spectrum00038-24-s0001.tif).** Capsule formation, agar plates.
**Supplemental material (Spectrum00038-24-s0002.xlsx).** Raw data; Fig S2 to S6.

### Open Peer Review

**PEER REVIEW HISTORY (review-history.pdf).** An accounting of the reviewer comments and feedback.

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
