## [Reviewer comments · Microbiology Spectrum]

Microbiology Spectrum

Sirtulin-Ypk1 regulation axis governs the TOR signaling pathway and fungal pathogenicity in *Cryptococcus neoformans*.

Zhenghua Chai, Yanjian Li, Jing Zhang, Chen Ding, Xiujuan Tong, and Zhijie Zhang

Corresponding Author(s): Zhijie Zhang, Department of Clinical Laboratory, Shengjing Hospital of China Medical University, Shenyang, China

Review Timeline:

Submission Date:	January 7, 2024
Editorial Decision:	February 18, 2024
Revision Received:	April 19, 2024
Accepted:	May 6, 2024

Editor: Alexandre Alanio

Reviewer(s): The reviewers have opted to remain anonymous.

Transaction Report:

DOI: <https://doi.org/10.1128/spectrum.00038-24>

Re: Spectrum00038-24 (**Sirtulin-Ypk1 regulation axis governs the TOR signaling pathway and fungal pathogenicity in *Cryptococcus neoformans*.**)

Dear Dr. Zhijie Zhang:

Thank you for the privilege of reviewing your work. Below you will find my comments, instructions from the Spectrum editorial office, and the reviewer comments.

Apart from the very pertinent comments from the reviewers, the authors should describe better their methods in the dedicated section.

Revision Guidelines

Sincerely,
Alexandre Alanio
Editor
Microbiology Spectrum

Reviewer #1 (Comments for the Author):

In this paper, Chai et al explore the impact of acetylation on Ypk1, a key component of the TOR signaling pathway. Authors build on their previous finding that Ypk1 is modified at two places and these acetylation modifications are crucial for its function. Overall, the experiments are well conducted and the conclusions are well supported by their experiments. I have a few

suggestions that will further improve the quality of the manuscript and strengthen the manuscript.

1. The authors show the difference in capsule in the Ypk1 mutants but do not include any other phenotypic characterization of the mutants. It will be important to have all other necessary phenotypes in the paper, mainly the ones that are directly relevant for TOR1 signaling such as nutrient limitation conditions. Furthermore, the images for capsule formation defects should be presented in the manuscript.
2. The authors mention additional quantifiable tests were done in liquid cell cultures but no further details are provided. These methods need to be described properly in the methods section.
3. In their growth experiments, the YPK1R mutant always exhibits more robust growth than the YPK1WT strains in both their spotting and liquid cultures experiments. What are the possible reasons and implications for this phenotype? These should be discussed in the revised manuscript.
4. The authors generated double mutants for *dac1*, 7, and 9 combinations. However, they do not discuss why only these three mutants were focused on for the generation of double mutants. Additionally, what would be the outcome in the triple mutant for *dac1*, *dac7* and *dac9*?
5. How do the double mutants for *dac1* and *dac7* behave in a capsule formation assay?
6. The presentation of their statistical analysis is rather confusing. In their figure legends, the authors mention *-based annotations whereas they write actual p-values in their figures. I would suggest they resort to a single consensus approach and describe their results.

Reviewer #2 (Comments for the Author):

Chai Z, et al. investigate the impact of acetylation of two lysin residues from the Ypk1 protein on growth and pathogenicity of *C. neoformans*. Ypk1 has been previously shown to be involved in pathogenicity of *C. neoformans* and its role in TOR signaling has been predicted based on other fungal models. The same group has shown previously that Ypk1 is acetylated on two lysins, K315, and K502. Here the Authors demonstrate that the mutant, in which the two K residues of Ypk1 are changed to Q to mimic acetylation, has attenuated virulence in the murine model of cryptococcosis. The same mutant has a mild growth defect in the presence of TOR inhibitor rapamycin, pointing to relevance of this modification to TOR pathway. Furthermore, rapamycin diminishes acetylation of Ypk1. The Authors provide evidence that two sirtuins, *Dac1* and *Dac7*, redundantly act as deacetylases towards Ypk1 and interact with Ypk1. While some of the effects are rather mild, this study provides cohesive evidence that *Dac1* and *Dac7* act as deacetylases towards Ypk1 and this deacetylation contributes to virulence of *C. neoformans*. Below are specific comments that should help to further improve this manuscript.

1. Ln 44: the sentence is unclear - "acetylation that reciprocally modulates" - that means that the mutant cannot be deacetylated (is permanently acetylated)?
2. Ln 45 "in response... to capsule formation, and fungal pathogenesis"? this does not sound correct.
3. Ln 49: it is not clear what "via protein-protein interaction" means? does this mean *Dac1* and *Dac7* need to interact with each other or each needs to interact with Ypk1?
4. Ln 107: I would not call them "virulence factor orthologs" as capsule for instance is a virulence factor. Perhaps the Authors could say "orthologs of proteins involved in virulence"?
5. It is not entirely clear from the Introduction and the Abstract what is new versus what has been published. I suggest that the Authors modify the text to make sure this is clear.
6. Ln 134: K315 not K305?
7. Ln 159: "the animal..." instead of "these animal"
8. Figure 2: Capsule thickness - relative % of total capsule thickness? sounds odd and is inconsistent with Y axis in Fig2A
9. The effect of double Q mutation on growth in the presence of rapamycin is very mild
10. Fig 3A,B - how was cell growth measured? was this OD? what were the cell densities?
11. Ln 241: Based Fig4, it looks like nine deletion strains were tested.
12. Ln 266: "This suggested that multiple..."
13. Ln 268-269: I would put full stop after "acetylation level of Ypk1." then say: "Simultaneously disrupting *DAC1* and *DAC7* led to enhanced acetylation"
14. Ln 270: growth phenotype of what (?) - the Q mutant.
15. In Figure 3, the Authors show growth as relative growth (?) but in Figure 5, the Authors show growth as OD600. Both show consistent (statistically significant) yet rather mild phenotypes.
16. Strain generation in Materials and Methods lacks critical information: The R and Q mutants were provided by Ding's Lab - Reference is needed if published or detailed description how they were made. What method was used for transformations? What is pXL-HYG plasmid? - reference is needed.
17. the Authors provide primer numbers, but no table is included with the corresponding sequences.
18. Animal Infection procedures lack details - they should be provided or reference to publication that has those details provided.
19. Ln 431: Capsule protocol - "as described elsewhere " - reference is needed. the description of how the relative capsule thickness was calculated is needed.
20. Ln 439: as described elsewhere - Reference is needed.
21. Ln 444: "Protein acetylation was detected after washing the beads four times with TBS buffer " this does not sound correct - after washing the sample was run on the electrophoresis gel and then western blot was performed, is that correct?

22. Ln 446-447 - similar to previous comment - it sounds like the protein-protein interactions were detected directly after washing.

23. Ln 453 - reference is missing.

Response to reviewer comments:

General response: We thank the reviewers and the editor for taking the time to review our manuscript. We sincerely appreciate your thoughtful feedback and constructive suggestions, which have undoubtedly improved the quality of our work. Based on your suggestion and request, we have made corrected modifications on the revised manuscript. And more detailed methods and steps have been modified. We hope that our work can be improved again. Furthermore, we would like to show the details as follows:

Reviewer #1 (Comments for the Author):

In this paper, Chai et al explore the impact of acetylation on Ypk1, a key component of the TOR signaling pathway. Authors build on their previous finding that Ypk1 is modified at two places and these acetylation modifications are crucial for its function. Overall, the experiments are well conducted and the conclusions are well supported by their experiments. I have a few suggestions that will further improve the quality of the manuscript and strengthen the manuscript.

Response: Thank you very much for your time involved in reviewing the manuscript and your very encouraging comments on the merits.

1. The authors show the difference in capsule in the Ypk1 mutants but do not include any other phenotypic characterization of the mutants. It will be important to have all other necessary phenotypes in the paper, mainly the ones that are directly relevant for TOR1 signaling such as nutrient limitation conditions. Furthermore, the images for capsule formation defects should be presented in the manuscript.

Response: We very much appreciate the comment from this expert. This is a very good suggestion, indeed it should be tested in the research. TOR1 signaling dynamically responds to nutrient limitation conditions to promote cell survival and maintain cellular homeostasis. During amino acid starvation, TOR operates independently as a positive regulator of autophagy through the conserved TORC2 and its downstream target kinase Ypk1(1). In the revised manuscript, we analyzed the starvation tolerance of these mutants under nitrogen starvation (SD-N) and carbon starvation (SD-C) (2). In the 7-day nitrogen starvation treatment, the *ypk1Δ* mutant exhibited pronounced growth defects, as expected. Meanwhile, *YPK1Q* exhibited growth defects almost identical to *ypk1Δ*. The same phenomenon was also observed during the 14-day carbon starvation treatment (Figure 3F). This phenomenon suggests that the acetylation modification level of Ypk1 plays a crucial regulatory role in responding to starvation tolerance conditions, and it impacts the process of autophagy. And it plays a more significant role in tolerance to nitrogen-only starvation than in to carbon-only starvation. Please see line 185-195 of the “Marked-Up Manuscript” document.

Additionally, the images for capsule formation are presented below and are supplemented in Supplemental Figure 1A. Thanks for the reviewer for pointing this out.

Figure 3F

Supplemental Figure 1A

2. The authors mention additional quantifiable tests were done in liquid cell cultures but no further details are provided. These methods need to be described properly in the methods section.

Response: We appreciate the comment and apologize for not clarifying the methods. The detailed steps have been supplemented in the Materials and Methods section. In summary, We utilizing 96-well microtitre plates to assess the growth of cells in liquid culture media. Overnight YPD cultures were washed three times with phosphate-buffered saline (PBS) and diluted to an optical density of 0.02 at 600 nm in fresh YPD, supplemented with 2 ng/ml rapamycin. Following this, 100 μl of the resulting cell suspension was carefully dispensed into individual wells of a 96-well plate. The well plate was subjected to incubation at a temperature of 30°C for either 12, 24 or 48 hr. Subsequently, optical density measurements at a wavelength of 600 nm were obtained using a Synergy HTX microplate reader manufactured by BioTek. The growth of the relevant strain was standardized by normalizing it to the well without rapamycin treatment. Six or twelve biological replicates were conducted for each strain. The data were graphed utilizing GraphPad Prism software. Two tails unpaired t-tests were used. Please see line 391-403 of the “Marked-Up Manuscript” document.

3. In their growth experiments, the *YPK1R* mutant always exhibits more robust growth than the *YPK1WT* strains in both their spotting and liquid cultures experiments. What are the possible reasons and implications for this phenotype? These should be discussed in the revised manuscript.

Response: The acetylation level of proteins is a dynamic process during cell growth, regulated by both acetyltransferases and deacetylases. In our data shows that the acetylation modification level of the Ypk1 protein rapidly decreases under stimulation with DNEM+10% FBS or rapamycin (Figure 2B, Figure 3E). Furthermore, the *YPK1WT* strain exhibits a very low levels of acetylation modification, indicating that Ypk1 protein exists in a non-acetylated state during the growth of *Cryptococcus neoformans*. Ypk1 will remain in a non-acetylated state when we artificially mutate acetylation sites K^{ac315} and K^{ac502} to arginine. It may be more conducive to its function within *YPK1R* strain. Studies have extensively shown that the non-acetylation state of proteins

contributes to the activation of protein functions. For example, *Cryptococcus neoformans* ISW1, an chromatin remodeling factor, when mutated in its *ISWIR* form, activates the protein ubiquitination pathway, contributing to the development of antifungal drug resistance(3). The K642R mutation in the mammalian mitochondrial fission protein dynamin-related protein 1 (Drp1) blocked palmitate-induced Drp1 phosphorylation, oligomerization, and activity in adult cardiomyocytes. This ultimately reduced cardiomyocyte death and heart dysfunction(4). In *Saccharomyces cerevisiae*, Ypk1 protein post-translational modification (PTM), phosphorylation, acts as a critical modulator for its protein activity and function(5). Perhaps the phosphorylation activity of Ypk1 is higher in its non-acetylated state. To further elucidate the interplay between these two modifications, a detailed analysis of Ypk1 phosphorylation and the associated target proteins must be performed using mass spectrometry. And these hypotheses require more in-depth research to be confirmed in the future. Please see Discussion section, line 277-283 of the “Marked-Up Manuscript” document.

4. The authors generated double mutants for *dac1*, 7, and 9 combinations. However, they do not discuss why only these three mutants were focused on for the generation of double mutants. Additionally, what would be the outcome in the triple mutant for *dac1*, *dac7* and *dac9*?

Response: We appreciate the comment and apologize for not clarifying the rationale of *dac1Δ*, *dac7Δ*, and *dac9Δ*. *Li et al.* have confirmed that there are two main deacetylase families present in *Cryptococcus neoformans*: the HDAC (Histone Deacetylase) family and sirtuin family(6). The HDAC family includes *Dac2–6*, 8, and 11, while the sirtuin family includes *Dac1*, 7, and 9. In this study, we found that the sirtuin deacetylase family regulates the acetylation of Ypk1 (data from Figure 4 and Figure 5). Moreover, the acetylation modification level of Ypk1 could only be increased when both *Dac1* and *Dac7* were deleted. However, whether it was a single knockout of *Dac9* or a double knockout in combination with *Dac1* or *Dac7*, the acetylation modification level of Ypk1 did not change. It is ruled out that *Dac9* is a deacetylase of Ypk1. Therefore, in the subsequent experiments, we focus on the interaction between *Dac1*, *Dac7*, and Ypk1, rather than on the triple mutant for *Dac1*, *Dac7* and *Dac9*. In the revised manuscript, we have provided a more comprehensive explanation for this section. Please see line 199-201 of the “Marked-Up Manuscript” document.

5. How do the double mutants for *dac1Δ* and *dac7Δ* behave in a capsule formation assay?

Response: According to your suggestion, we tested the capsule structure formation ability of *dac1Δ* and *dac7Δ* and *dac1Δdac7Δ* strains. The results showed a significant reduction in the thickness of the capsule structure in all three mutant strains, especially in the *dac1Δdac7Δ* double knockout strain. It is consistent with the phenotype of the *YPK1Q* strain's capsule structure shown in Figure 2A. However, the *dac1Δdac7Δ* double knockout exhibited a more severe defect, indicating the importance of downstream genes regulated by *Dac1* and *Dac7*, which warrants further investigation in the future (Figure.5C). The images for capsule formation are presented below and are supplemented in Supplemental Figure 1B.

Meanwhile, we supplemented the experiments on the nutrient starvation tolerance and rapamycin resistance of single knockout strain *dac1Δ* and *dac7Δ*, as well as the *dac1Δdac7Δ* double knockout

strain. From the results, it can be seen that under YPD conditions, the growth of *dac1Δ* and *dac7Δ* and *dac1Δdac7Δ* strains was consistent with the wild-type H99 strain. However, after 7-days nitrogen starvation treatment and 14-days carbon starvation treatment, the *dac7Δ* single knockout strain did not show any growth defects, while the *dac1Δ* exhibited slight growth defects, and the *dac1Δdac7Δ* double knockout strain showed significant growth defects (Figure.5D). In the rapamycin resistance experiment, only the *dac1Δdac7Δ* double knockout strain showed slight growth defects (Supplemental Figure 1C). This phenomenon confirms the functional redundancy of Dac1 and Dac7, indicating that both collectively regulate the TOR pathway and nutrient intake. The deacetylase also positively regulates the process of autophagy, and its absence affects the growth of strains under nutritional stress conditions. Please see line 217-228 of the “Marked-Up Manuscript” document.

Figure 5C

Figure 5D

Supplemental Figure 1B

Supplemental Figure 1C

6. The presentation of their statistical analysis is rather confusing. In their figure legends, the authors mention *-based annotations whereas they write actual p-values in their figures. I would suggest they resort to a single consensus approach and describe their results.

Response: We thank for the comment, and apologize for the confusion in the original manuscript. In the revised manuscript, we use actual *p*-values consistently for representation. Detailed statistical methods and descriptions will be provided in the Materials and Methods section. Please see line 438-441 of the “Marked-Up Manuscript” document.

Reviewer #2 (Comments for the Author):

Chai Z, et al. investigate the impact of acetylation of two lysin residues from the Ypk1 protein on growth and pathogenicity of *C. neoformans*. Ypk1 has been previously shown to be involved in pathogenicity of *C. neoformans* and its role in TOR signaling has been predicted based on other fungal models. The same group has shown previously that Ypk1 is acetylated on two lysins, K315, and K502. Here the Authors demonstrate that the mutant, in which the two K residues of Ypk1 are changed to Q to mimic acetylation, has attenuated virulence in the murine model of cryptococcosis. The same mutant has a mild growth defect in the presence of TOR inhibitor rapamycin, pointing to relevance of this modification to TOR pathway. Furthermore, rapamycin diminishes acetylation of Ypk1. The Authors provide evidence that two sirtuins, Dac1 and Dac7, redundantly act as deacetylases towards Ypk1 and interact with Ypk1. While some of the effects are rather mild, this study provides cohesive evidence that Dac1 and Dac7 act as deacetylases towards Ypk1 and this deacetylation contributes to virulence of *C. neoformans*. Below are specific comments that should help to further improve this manuscript.

Response: We thank reviewer for giving us many valuable comments to improve our manuscript. We carefully considered all comments.

1. Ln 44: the sentence is unclear - "acetylation that reciprocally modulates" - that means that the mutant cannot be deacetylated (is permanently acetylated)?

Response: We appreciate the comments. Recombinant KQ mutant (lysine residues are substituted with glutamine as a mimic of acetyl lysine), and KR mutant (lysine residues are substituted with arginine as a mimic of nonacetylated lysine), are widely used to study the effects of acetylation. In Line 44, we characterized the impact of mimic acetylation modification on protein function by constructing the *YPK1Q* mutant strain. This implies that *YPK1Q* will be permanently acetylated at those Lys residues and cannot be deacetylated.

2. Ln 45 "in response... to capsule formation, and fungal pathogenesis"? this does not sound correct.

Response: We appreciate the comments and the manuscript has been corrected. Please see line 39-42 of the "Marked-Up Manuscript" document.

3. Ln 49: it is not clear what "via protein-protein interaction" means? does this mean Dac1 and Dac7 need to interact with each other or each needs to interact with Ypk1?

Response: We thank for the comment, and apologize for the confusion in the original manuscript. Indeed, Dac1 and Dac7 directly interact with Ypk1 to facilitate the deacetylation modification process. The manuscript text has been modified accordingly. Please see line 44-46 of the "Marked-Up Manuscript" document.

4. Ln 107: I would not call them "virulence factor orthologs" as capsule for instance is a virulence factor. Perhaps the Authors could say "orthologs of proteins involved in virulence"?

Response: We thank for the comment and the manuscript has been corrected. Please see line 106 of the “Marked-Up Manuscript” document.

5. It is not entirely clear from the Introduction and the Abstract what is new versus what has been published. I suggest that the Authors modify the text to make sure this is clear.

Response: We thank for the comment and the manuscript has been corrected. Please see the Abstract and the Introduction.

6. Ln 134: K315 not K305?

Response: We thank for the comment and the manuscript has been corrected. Please see line 125 of the “Marked-Up Manuscript” document.

7. Ln 159: "the animal..." instead of "these animal"

Response: We thank for the comment and the manuscript has been corrected. Please see line 142-144 of the “Marked-Up Manuscript” document.

8. Figure 2: Capsule thickness - relative % of total capsule thickness? sounds odd and is inconsistent with Y axis in Fig2A

Response: We appreciate the comments. The capsule thickness of the relevant strain was standardized by normalizing it to *YPK1WT* strain. The Y-axis represents the relative capsule structure thickness compared to the *YPK1WT* strain. We have provided a revised description for the Y-axis. Please see Figure 2A.

9. The effect of double Q mutation on growth in the presence of rapamycin is very mild

Response: We absolutely agree with the comment from this expert. Indeed, we observed a slight growth defect of the *YPK1Q* strain compared to the *YPK1WT* strain on agar plates. Since *YPK1WT* represents an intermediate acetylation state, when comparing to the extremes of Ypk1 acetylation, namely the non-acetylated state *YPK1R* and the fully acetylated state *YPK1Q*, we observed significant differences in their growth patterns. This was consistent even in liquid culture conditions. Therefore, this can explain the acetylation status of Ypk1 affects rapamycin resistance, reflecting one of the functions of the Ypk1 protein.

10. Fig 3A,B - how was cell growth measured? was this OD? what were the cell densities?

Response: We appreciate the comment and apologize for not clarifying the methods. The detailed steps have been supplemented in the Materials and Methods section. In brief, we utilizing 96-well microtitre plates to assess the growth of cells in liquid culture medium. Overnight YPD cultures were washed three times with phosphate-buffered saline (PBS) and diluted to an optical density of 0.02 at 600 nm in fresh YPD, supplemented with 2 ng/ml rapamycin. Following this, 100 µl of the

resulting cell suspension was carefully dispensed into individual wells of a 96-well plate. The well plate was subjected to incubation at a temperature of 30°C for either 12, 24 or 48 hr. Subsequently, optical density measurements at a wavelength of 600 nm were obtained using a Synergy HTX microplate reader manufactured by BioTek. The growth of the relevant strain was standardized by normalizing it to the well without rapamycin treatment. Six or twelve biological replicates were conducted for each strain. The data were graphed utilizing GraphPad Prism software. Two tails unpaired t-tests were used. Please see line 391-403 of the “Marked-Up Manuscript” document.

11. Ln 241: Based Fig4, it looks like nine deletion strains were tested.

Response: We thank the reviewer for pointing this out. The manuscript has been corrected. Please see line 208 of the “Marked-Up Manuscript” document.

12. Ln 266: "This suggested that multiple..."

Response: The manuscript has been corrected. Please see line 212 of the “Marked-Up Manuscript” document.

13. Ln 268-269: I would put full stop after "acetylation level of Ypk1." then say: "Simultaneously disrupting DAC1 and DAC7 led to enhanced acetylation"

Response: The manuscript has been corrected. Please see line 215 of the “Marked-Up Manuscript” document.

14. Ln 270: growth phenotype of what (?) - the Q mutant.

Response: We thank the reviewer for pointing this out. The manuscript has been corrected. Please see line 221 of the “Marked-Up Manuscript” document.

15. In Figure 3, the Authors show growth as relative growth (?) but in Figure 5, the Authors show growth as OD600. Both show consistent (statistically significant) yet rather mild phenotypes.

Response: We thank the reviewer for pointing this out and apologize for the confusion in the original manuscript. In the revised manuscript, we have standardized the axis representation for the data in the liquid culture experiments. Please see Figure. 3B, Figure. 5B.

16. Strain generation in Materials and Methods lacks critical information: The R and Q mutants were provided by Ding's Lab - Reference is needed if published or detailed description how they were made. What method was used for transformations? What is pXL-HYG plasmid? - reference is needed.

Response: We thank the reviewer for pointing this out and apologize for the confusion in the original manuscript. The detailed construction methods, plasmids used, and references have been

added to the manuscript. Please see Materials and Methods from line 337 to 339, 349-350 of the “Marked-Up Manuscript” document.

17. The Authors provide primer numbers, but no table is included with the corresponding sequences.

Response: We thank the reviewer for pointing this out. The primer list has been included in the Supplementary file 1. Meanwhile, the strain list is also supplemented in the Supplementary file 1.

18. Animal Infection procedures lack details - they should be provided or reference to publication that has those details provided.

Response: We thank the reviewer for pointing this out. The more detailed animal infection procedures have been described in the Materials and Methods. Please see line 369-380 of the “Marked-Up Manuscript” document.

19. Ln 431: Capsule protocol - "as described elsewhere " - reference is needed. the description of how the relative capsule thickness was calculated is needed.

Response: We thank the reviewer for pointing this out. The references have been added to the revised manuscript. We normalize the data using the thickness of *YPK1WT* strain (in Figure 2A) and H99 (in Figure 5C) as the standard. Please see line 381-390 of the “Marked-Up Manuscript” document.

20. Ln 439: as described elsewhere - Reference is needed.

Response: We thank the reviewer for pointing this out. The references have been added to the revised manuscript. Please see line 407 of the “Marked-Up Manuscript” document.

21. Ln 444: "Protein acetylation was detected after washing the beads four times with TBS buffer " this does not sound correct - after washing the sample was run on the electrophoresis gel and then western blot was performed, is that correct?

Response: Thank you for pointing this out. For the detection of protein acetylation, after incubating the protein samples with Anti-FLAG M2 magnetic beads for 4 hours, we washed them four times with TBS (50 mM Tris-HCl, 150 mM NaCl, 1% Triton X-100, pH 7.4) supplemented with 3 μ M TSA and 20 mM NAM. Then, the bound proteins were extracted into protein loading buffer at 95°C for 5 min, followed by Western blotting. Monoclonal and polyclonal Kac (1:2500, PTM Bio) was used to detect the level of protein acetylation. The signal was captured using a ChemiDoc XRS+ (Bio-Rad). More detailed methods and steps have been modified in the manuscript. Please see line 411-417 of the “Marked-Up Manuscript” document.

22. Ln 446-447 - similar to previous comment - it sounds like the protein-protein interactions were detected directly after washing.

Response: Thank you for pointing this out. For protein-protein interactions were detected after washing with TBS buffer without Triton X-100. The bound proteins were extracted into protein loading buffer at 95°C for 5 min, followed by Western blotting. Anti-M2 Flag (1:5000, Abcam), anti-HA (1:5000, Abcam), anti-Histone H3 (1:5000, Cell Signaling Technology) were used for detection the target protein. The signal was captured using a ChemiDoc XRS+ (Bio-Rad). More detailed methods and steps have been modified in the manuscript. Please see line 417-422 of the “Marked-Up Manuscript” document.

23. Ln 453 - reference is missing.

Response: We thank the reviewer for pointing this out. The references have been added to the revised manuscript. Please see line 424 of the “Marked-Up Manuscript” document.

1. Vlahakis A, Graef M, Nunnari J, Powers T. 2014. TOR complex 2-Ypk1 signaling is an essential positive regulator of the general amino acid control response and autophagy. *Proceedings of the National Academy of Sciences of the United States of America* 111:10586-10591.
2. Zhao X, Feng W, Zhu X, Li C, Ma X, Li X, Zhu X, Wei D. 2019. Conserved Autophagy Pathway Contributes to Stress Tolerance and Virulence and Differentially Controls Autophagic Flux Upon Nutrient Starvation in *Cryptococcus neoformans*. *Front Microbiol* 10:2690.
3. Meng Y, Ni Y, Li Z, Jiang T, Sun T, Li Y, Gao X, Li H, Suo C, Li C, Yang S, Lan T, Liao G, Liu T, Wang P, Ding C. 2024. Interplay between acetylation and ubiquitination of imitation switch chromatin remodeler Isw1 confers multidrug resistance in *Cryptococcus neoformans*. *eLife* 13:e85728.
4. Hu Q, Zhang H, Gutierrez Cortes N, Wu D, Wang P, Zhang J, Mattison JA, Smith E, Bettcher LF, Wang M, Lakatta EG, Sheu SS, Wang W. 2020. Increased Drp1 Acetylation by Lipid Overload Induces Cardiomyocyte Death and Heart Dysfunction. *Circ Res* 126:456-470.
5. Roelants FM, Breslow DK, Muir A, Weissman JS, Thorner J. 2011. Protein kinase Ypk1 phosphorylates regulatory proteins Orm1 and Orm2 to control sphingolipid homeostasis in *Saccharomyces cerevisiae*. *Proc Natl Acad Sci U S A* 108:19222-7.
6. Li YJ, Li HL, Sui MF, Li MH, Wang JM, Meng Y, Sun TS, Liang QJ, Suo CH, Gao XD, Li C, Li ZR, Du W, Zhang BH, Sai SX, Zhang Z, Ye J, Wang HC, Yue S, Li JY, Zhong ML, Chen CB, Qi SL, Lu L, Li DC, Ding C. 2019. Fungal acetylome comparative analysis identifies an essential role of acetylation in human fungal pathogen virulence. *Communications Biology* 2.

Re: Spectrum00038-24R1 (**Sirtulin-Ypk1 regulation axis governs the TOR signaling pathway and fungal pathogenicity in *Cryptococcus neoformans*.**)

Dear Dr. Zhijie Zhang:

Your manuscript has been accepted, and I am forwarding it to the ASM production staff for publication. Your paper will first be checked to make sure all elements meet the technical requirements. ASM staff will contact you if anything needs to be revised before copyediting and production can begin. Otherwise, you will be notified when your proofs are ready to be viewed.

Sincerely,
Alexandre Alanio
Editor
Microbiology Spectrum